



# Global, Satellite-Driven Estimates of Heterotrophic Respiration

Alexandra G. Konings[1], A. Anthony Bloom[2], Junjie Liu[2], Nicholas C. Parazoo[2], David S. Schimel[2], Kevin W. Bowman[2]

[1]Department of Earth System Science, Stanford University, Stanford, CA 94305, USA

[2]NASA Jet Propulsion Laboratory, California Institute of Technology, Pasadena, CA 91109, USA

*Correspondence to*: Alexandra G. Konings (konings@stanford.edu)

**Abstract.** While heterotrophic respiration ($R_h$) makes up about a quarter of gross global terrestrial carbon fluxes, it remains among the least observed carbon fluxes, particularly outside the mid-latitudes. In situ measurements collected in the Soil Respiration Database (SRDB) number only a few hundred worldwide. Similarly, only a single data-driven wall-to-wall

estimate of annual average heterotrophic respiration exists, based on bottom-up upscaling of SRDB measurements using an assumed functional form to account for climate variability. In this study, we exploit recent advances in remote sensing of terrestrial carbon fluxes to estimate global variations in heterotrophic respiration in a top-down fashion at monthly temporal resolution and 4x5º spatial resolution. We combine net ecosystem productivity estimates from atmospheric inversions of the NASA Carbon Monitoring System- Flux (CMS-Flux) with an optimally-scaled gross primary productivity dataset based on

satellite-observed solar-induced fluorescence variations to estimate total ecosystem respiration as a residual of the terrestrial carbon balance. The ecosystem respiration is then separated into autotrophic and heterotrophic components based on a spatially-varying carbon use efficiency retrieved in a model-data fusion framework (the CARbon DAta MOdel fraMework, CARDAMOM). The resulting dataset is independent of any assumptions about how heterotrophic respiration responds to climate or substrate variations. It estimates an annual average global average heterotrophic respiration flux of 43.6 ± 19.3 Pg

C/yr. These top-down estimates are compared to bottom-up estimates of annual heterotrophic respiration, with new uncertainty estimates that partially account for sampling and model errors. Top-down heterotrophic respiration estimates are higher than those from bottom-up upscaling everywhere except at high latitudes, and are 30% greater overall (43.6 Pg C/yr vs. 33.4 Pg C/yr). The uncertainty ranges of both methods are comparable, except poleward of 45 degrees North, where bottom-up uncertainties are greater. The ratio of top-down heterotrophic to total ecosystem respiration varies seasonally by as much as

0.6 depending on season and climate, illustrating the importance of studying the drivers of autotrophic and heterotrophic respiration separately, and thus the importance of data-driven estimates of $R_h$ such as those estimated here.

## 1 Introduction

The terrestrial carbon cycle-climate feedback (together with atmospheric processes) is a dominant contributor to the uncertainty of temperature projections in 2100 (Booth et al., 2012). The future effect of carbon-climate feedbacks depends on

the climate sensitivity of net terrestrial carbon fluxes, which are a close balance of net primary productivity, disturbance-related

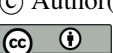


fluxes, and heterotrophic respiration ($R_h$). The overall sensitivity of the terrestrial carbon uptake is thus dependent on the climatic response of these fluxes. Model-based estimates of global $R_h$ vary by almost 50% and are highly uncertain (Shao et al., 2013), especially in the tropics (Tian et al., 2015). The climatic sensitivity of $R_h$ is also the primary driver of the large divergence across modeled global soil carbon pools (Tian et al., 2015; Todd-Brown et al., 2013), the largest terrestrial carbon

pool (Jobbágy and Jackson, 2000).

    Few in situ measurements exist to constrain $R_h$, particularly in the tropics (Xu et al., 2016). For example, the international Soil Respiration DataBase (SRDB), which aims to compile data from all published studies of soil and heterotrophic respiration (Bond-Lamberty and Thomson, 2010), includes only 21 sites with $R_h$ information in Central and South America, and only 2 in Africa. The highly limited number of $R_h$ data is likely affected by the relative difficulty and

uncertainty of methods for partitioning total soil respiration ($R_s$) fluxes – which can be easily measured using respiration chambers – into autotrophic and heterotrophic components. Performing this partitioning requires isotopic measurements or destructive techniques such as girdling or trenching (Ryan and Law, 2005). Like $R_s$, total ecosystem respiration $R_{eco}$ is also often considered as a counter-part to photosynthetic fluxes, but is rarely partitioned further.  However, because most carbon cycle and ecosystem models represent autotrophic and heterotrophic components separately, and because the climatic and soil

sensitivities of the autotrophic and heterotrophic components of soil respiration differ (Metcalfe et al., 2010; Scott-Denton et al., 2006), it is challenging to translate soil or ecosystem respiration data to improvements of model representations for $R_h$. Although meta-analyses using data such as the SRDB have been used to understand the sources of spatial variability in soil respiration (Hursh et al., 2016) and heterotrophic respiration (Shao et al., 2013) rates, such studies are limited (by data availability) to consideration of annual respiration fluxes and sparse, discrete points in space. Thus, while GPP is highly

uncertain (Anav et al., 2015), it is far more constrained by observations than respiration, especially $R_h$, which must be considered among the most uncertain fluxes in the carbon cycle. Temporally variable and spatially extensive estimates of $R_h$ are therefore needed to better understand its drivers.

    Starting several decades ago with Raich & Schlesinger (1992), several authors have tried to upscale sparse measurements to estimate global $R_s$. Most commonly, this is performed using a spatially explicit exponential model of the

relationship between $R_s$ and temperature, modified by land cover (Adachi et al., 2017), soil properties (Chen et al., 2013), or soil moisture (Xu et al., 2016) limitations. Recent papers have also used machine learning methods to upscale the relationship between $R_s$ and climate and biogeophysical properties, including random forest models (Jian et al., 2018) and artificial neural networks (Zhao et al., 2017). However, no similar effort has been made for estimating global $R_h$. Only Hashimoto et al. (2015) have extended a rigorous estimate of global, upscaled $R_s$ to an estimate of global $R_h$. This was achieved by employing a

previously noted apparent relationship between annual $R_s$ and $R_h$ at a given site (Bond-Lamberty et al., 2004). However, the Hashimoto et al approach assumes a specific functional form for the relationship between climate and $R_s$ (and thus, $R_h$), so that investigations of climatic sensitivities of $R_h$ with this dataset are potentially circular. Furthermore, the approach assumes




that base respiration rates and sensitivity parameters for temperature and precipitation to soil moisture are constant across the globe. This approach therefore cannot account for known dependencies of heterotrophic respiration on microbial biomass and composition (Johnston and Sibly, 2018; Walker et al., 2018; Wieder et al., 2013; Zhou et al., 2011) and substrate type (Cornwell et al., 2008). Modelling $R_h$ as a function of precipitation alone is also inconsistent with theoretical, laboratory, and field studies

that have found $R_h$ to be a function of soil water potential (Manzoni et al., 2012; Moyano et al., 2012, 2013), which is non-linearly related to precipitation depending on soil properties, vegetation cover, topography, and more.

In this paper, we introduce an alternative approach to estimating $R_h$ at global or regional scales using remote sensing. Rather than a bottom-up approach to aggregating sparse point-based measurements, we propose a 'top-down' method that naturally captures average values over large scales. The method derives $R_h$ as the residual of satellite-constrained estimates of

the carbon balance: net ecosystem productivity (NEP), gross primary productivity (GPP) and $R_a$. The NEP (the net difference between photosynthetic and respiration fluxes: NEP = GPP - $R_a$ - $R_h$) is based on atmospheric inversions of satellite observations of column $xCO_2$ and $xCO$, and GPP is based on upscaling solar-induced fluorescence (SIF). The $R_a$ is calculated based on GPP and carbon-use efficiency estimates from a remote-sensing constrained model-data fusion framework. The top-down approach is applied to the period 2010-2012. Coarse-resolution $R_h$ estimates are difficult to validate using in situ

measurements because of representativeness errors. Instead, we rigorously compare the top-down method and its uncertainties to those of bottom-up $R_h$ estimation, in this case as performed by Hashimoto et al. (2015).

## 2 Methods

### 2.1 Top-down Estimates

#### 2.1.1 Top-down approach

As summarized in Fig. 1, the top-down $R_h$ at grid-scale is calculated at the residual of observationally-constrained estimates of the carbon balance:

$$R_h = GPP - NEP - R_a, \qquad (1)$$

The combination of NEP and GPP allows calculation of the ecosystem respiration $R_{eco}$, but an estimate of $R_a$ is required to separate $R_{eco}$ into $R_a$ and $R_h$ components. The $R_a$ is calculated based on the GPP and on carbon-use efficiency (CUE). Specifically, the autotrophic respiration $R_a$ is assumed to be proportional to GPP according to a spatially variable but

temporally constant CUE, where CUE is defined as the ratio of net primary production (NPP) to GPP. Thus, CUE = 1 – $R_a$/GPP. Eq. (1) then becomes:

$$R_h = GPP - NEP - (1 - CUE)GPP, \qquad (2)$$

The CUE is commonly assumed constant at a given location (Gifford, 2003; McCree and Troughton, 1966), but has been found to vary depending on ecosystem type, stand age, and forest management (Collalti et al., 2018; Gifford, 2003; De Lucia et al., 2007). Note that by calculating autotrophic respiration as proportional to GPP, we classify the release of $CO_2$ from





decomposition of root exudates by mycorrhizal fungi (Trumbore, 2006) as autotrophic rather than heterotrophic respiration. This is arguably a misclassification, but is consistent with most in situ methods for measuring heterotrophic respiration (e.g. girdling, trenching, or isotopic measurements) (Ryan and Law, 2005).

The use of this method to calculate spatio-temporal variations in $R_h$ is enabled by the fact that estimates of each of NEP, GPP, and CUE are available that are based on remote sensing and data assimilation. These datasets are further discussed in the next Section.

### 2.1.2 Datasets used and implementation

The NEP is determined from an atmospheric inversion of remotely sensed columnar carbon dioxide and carbon monoxide observations in the CMS-Flux system. It is described in detail in Bowman et al. (2017) and (Liu et al., 2017), but summarized here for convenience. It has been validated using methods introduced in Liu and Bowman (2016). CMS-Flux estimates carbon fluxes through a 4D-variational inversion approach that ingests columnar $xCO_2$ observations from the Greenhouse gases Observing Satellite (GOSAT) and CO observations from the Measurement of Pollution in the Troposphere

Instrument (MOPITT) (Worden et al., 2010) into the GEOS-Chem atmospheric transport model and its adjoint (Bey et al., 2001; Henze et al., 2007; Nassar et al., 2010; Suntharalingam et al., 2004). The net fluxes are further decomposed into biomass burning, oceanographic, fossil fuel, chemical sources (including shipping, aviation, and others), and NEP components. The biomass burning emissions are constrained by the MOPITT CO observations and published $CO/CO_2$ ratios. Anthropogenic and oceanographic priors for the fluxes come from the Fossil Fuel Land Data Assimilation System (Asefi-Najafabady et al.,

2014; Rayner et al., 2010) and ECCO2-Darwin oceanographic model (Brix et al., 2015), respectively, and NEP flux priors come from the Carnegie-Ames-Stanford-Approach (CASA) model simulations. As shown in Fig. S1, the posterior and prior fluxes of NEP differ significantly almost everywhere – 42% of pixels have a normalized root-mean-square-difference between the prior and posterior fluxes greater than 1, consistent with a previous observing system simulation experiment for the CMS-Flux system (Liu et al., 2014). The uncertainty of the NEP estimates is assumed to be normally distributed with a spatially and

temporally varying standard deviation estimated in the atmospheric inversion via a Monte Carlo approach (Bousserez et al., 2015).

    The GPP is determined based on an optimal rescaling of SIF observations. SIF is a by-product of photosynthesis and therefore provides direct information about the magnitude of GPP (Porcar-Castell et al., 2014). The information content of SIF for photosynthesis has been demonstrated using field-scale measurements (Yang et al., 2015) and by comparing satellite-based

data to eddy-covariance towers (Guanter et al., 2014; Joiner et al., 2014; Sun et al., 2017b; Wood et al., 2017; Zuromski et al., 2018), carbon dioxide mole fractions in Amazonia (Parazoo et al., 2013), and machine-learning based estimates of GPP (Alemohammad et al., 2017). Despite the abundance of evidence that SIF carries information about GPP, the linear constant



of proportionality between SIF and GPP depends on the light use efficiency of the vegetation in question as well as the satellite efficiency at capturing photons and is difficult to estimate a priori. Here, we use GPP estimates from Parazoo et al. (2014), which used a Bayesian approach to determine an optimal seasonally and spatially varying scaling parameter between SIF and prior GPP along with explicit uncertainty estimates. Monthly GPP at each grid point is inferred from a precision-weighted

minimization of SIF, which is regressed against biome-specific GPP from upscaled flux tower data (Frankenberg et al., 2011; Jung et al., 2011), and prior GPP from eight terrestrial ecosystem models in the TRENDY project (Sitch et al., 2015). This approach has been used to examine regional GPP responses to climate variability and drought, and has been extensively validated against flux tower data (Bowman et al., 2017; Liu et al., 2017; Parazoo et al., 2014, 2015). The uncertainty of these GPP estimates is assumed to follow a normal distribution.

The CUE is determined from a 10-year (2001-2010) run of CARDAMOM (Bloom & Williams, 2015; Bloom et al., 2016), in which uncertainties are explicitly represented as probability density functions computed from an ensemble. CARDAMOM is a model-data fusion system that uses a Bayesian framework to determine global ecological parameter combinations that minimize the mismatch with observations while still satisfying a set of ecological realism and dynamic stability constraints to regularize the inversion. CARDAMOM is built on the underlying Data Assimilation Linked Ecosystem

Carbon model version 2, DALEC2 model (Bloom & Williams, 2015; Williams et al., 2005), with assimilation of observations of leaf area index, burned area, tropical biomass, and soil carbon (Bloom et al., 2016). Within CARDAMOM, a constant fraction $f_a$ of photosynthetic carbon gain is assumed to be allocated to autotrophic respiration (note that $f_a = 1-CUE$). The fraction $f_a$ is directly linked to the allocation fractions of photosynthetic carbon to other pools (labile, wood, foliar, and fine root carbon) through conservation of mass. The allocation fractions directly influence the observed quantities used for

CARDAMOM parameterization (e.g. LAI, tropical biomass, and soil carbon), and are subject to several ecological realism constraints. The resulting range of global CUE (between 0.35 and 0.6, shown in Fig. 2) is consistent with results found from meta-analyses (Gifford, 2003; De Lucia et al., 2007) and is also supported by theoretical considerations based on conservation of mass (Van Oijen et al., 2010) and plant carbon dynamics (Dewar et al., 1998). Average values of CARDAMOM CUE are generally lowest in the tropics, consistent with previous site-specific observations (Amthor, 2000; Chambers et al., 2004; De

Lucia et al., 2007; Piao et al., 2010).

Autotrophic respiration may depend on stored supplies of carbon, causing a decoupling between the seasonality of GPP and $R_a$ and thus temporal variation in CUE. This is particularly common in deciduous trees in mid- and high-latitudes (Epron et al., 2012; Kuptz et al., 2011). Less is known about seasonal variations of CUE in tropics. Although small variations in CUE (e.g. <= 0.05) have been observed in both highland and lowland Amazonian sites, these variations were found to be

small relative to seasonal variations in allocation rates to non-respiratory carbon pools (Doughty et al., 2015; Rowland et al., 2014). Nevertheless, the assumption of constant CUE likely adds error to the top-down estimates of $R_h$. This error is partially accounted for by the wide uncertainty range used for CUE. We further performed a sensitivity analysis in which the $R_h$ derived




using an assumption of constant CUE was compared to the $R_h$ with a systematic seasonal variability in CUE. Although little is known about the true temporal variation of CUE across the globe, we here assumed a seasonal cycle of CUE proportional to that of GPP, but re-normalized to have a mean equal to the constant CARDAMOM CUE and a standard deviation of 0.1 at each pixel. That is,

$$CUE(x, y, t) = CUE_{CARD}(x, y) + \frac{0.1}{std_t(GPP(x, yt) - \overline{GPP})}(GPP(x, y, t) - \overline{GPP}) ,\qquad(3)$$

where *(x,y)* determines a pixel location in space, *t* is the monthly time vector, $CUE_{CARD}(x,y)$ is the constant CUE determined from CARDAMOM, *std* refers to standard deviation, and the overbar denotes a time-average over the entire period. The use of a CUE proportional to GPP is chosen so as to provide a structure to the temporal variability of CUE that is potentially realistic for each pixel (i.e. not completely random), even if little is known about the overall controls on temporal variability

in CUE. The 0.1 standard deviation magnitude is fairly conservative unless true temporal variation in CUE is much larger than spatial variation – the spatial standard deviation of CARDAMOM CUE across all global land surfaces is 0.06.

When calculating $R_h$, in either the main or sensitivity analyses, all datasets are averaged to the same monthly temporal and 4° latitude x 5° longitude spatial resolution, the native resolution of flux estimates from CMS-Flux. We use NEP and GPP data over the period 2010-2012, when the CMS-Flux data have been validated in the most detail. However, even at 4° by 5°

resolution, the precision of CMS-Flux data can still be poor. To reduce error, all visual maps are presented after applying a 3 pixel by 3 pixel moving average smoother. When calculating $R_h$, a 4000-member ensemble is used for explicit simulation of the uncertainty distributions of each of the input variables. The NEP is a small number that is the balance of many larger components, so small errors in NEP could lead to large compensating errors in $R_h$. To reduce the effect of such compensating errors, a constraint on the signs of $R_h$, $R_a$, and GPP is used to ensure the estimated $R_h$ is physically realistic (Bloom and

Williams, 2015; Parazoo et al., 2018) – each of these three fluxes is required to be positive. A simple accept-reject sampling scheme is used that rejects ensemble members that violate this criterion. For each of these ensemble members, new samples of the uncertainty distribution of NEP, GPP, and CUE are drawn until each of $R_h$, $R_a$, and GPP for that ensemble member are positive. Using such a constraint is equivalent to using a Bayesian scheme with prior distributions for $R_h$, $R_a$, and GPP that are 0 for negative values and 1 otherwise.

The uncertainty of the resultant $R_h$ is a combination of the uncertainty in the three input datsets: NEP, GPP, and CUE. There is some non-linearity to this combination because the positive flux constraints limit what uncertainty combinations are considered to lead to acceptable $R_h$. To estimate how much each of these datasets contributes to the overall $R_h$ uncertainty, $R_h$ is re-estimated three times, but in each case only one of the input datasets is given non-zero uncertainty. The resultant magnitude of the uncertainty in $R_h$ is then compared.






### 2.2 Bottom-up estimates

#### 2.2.1 Approach

To the best of our knowledge, only Hashimoto et al. (2015) have previously estimated $R_h$ based on upscaling in situ measurements. Their method is based primarily on estimating $R_s$, for which a simple functional form adapted from Raich et al (2002) is used:

$$R_s = F \times e^{aT - bT^2} \times \frac{\alpha P_t + (1-\alpha)P_{t-1}}{K + \alpha P_t + (1-\alpha)P_{t-1}}, \tag{4}$$

where $F$ [gC m$^{-2}$ dy$^{-1}$] is a base rate, $a$ [$^\circ$C$^{-1}$] and $b$ [$^\circ$C$^{-2}$] control the sensitivity to temperature $T$ [$^\circ$C]. The $R_s$ also depends on the current-month precipitation $P_t$ [cm mo$^{-1}$] and the previous-month precipitation $P_{t-1}$ [C$^{-1}$], with the relative weight of each determined by $a$ [-]. The $K$ [cm mo$^{-1}$] parameter also controls the influence of precipitation. For lack of more information, all parameters are assumed to be global constant, so that the only spatial variation is provided by variations in the climatic drivers. Hashimoto et al (2015) used temperature and precipitation from the Climate Research (CRU) 3.21 (Harris et al., 2014) and fit the above function to observations from the SRDB using a Bayesian Markov Chain Monte Carlo scheme. To determine the annual average $R_h$ at a location based on annual $R_s$, Hashimoto et al. (2002) employed a previously determined relationship between annual and heterotrophic respiration (Bond-Lamberty et al., 2000):

$$\ln(R_h) = c + d\ln(R_s), \tag{5}$$

where $c = 1.22$ and $d = 0.73$. While it is in theory possible to apply Eq. (5) to any number of recent bottom-up $R_s$ estimation approaches, we here apply it only to the estimates from Hashimoto et al. (2015) in Eq. (4), both for consistency with the literature and since the data from Hashimoto et al. (2015) are among the most commonly used bottom-up $R_s$ estimates.

#### 2.2.2 Parametrization and Implementation

We implemented Eq. (4) using climate data from the CRU 4.01 and using the maximum a posteriori parameter values from Hashimoto et al. (2015) (that is, F = 1.68 gC m$^{-2}$ dy$^{-1}$, a = 0.0528 $^\circ$C$^{-1}$, b = 0.000628 $^\circ$C$^{-2}$, $\alpha$ = 0.98, and K = 1.20 cm mo$^{-1}$) to determine monthly resolution estimates of $R_s$. The $R_s$ estimates were then temporally aggregated to determine annual $R_h$ using Eq. (5). These are referred to as the bottom-up estimates below.

Extrapolating from a limited sample of parameters with multiple fitted parameters carries the risk of overfitting. Fortunately, several measurements of $R_s$ and $R_h$ have been added to the SRDB since the Hashimoto et al (2015) article; the number of $R_s$ measurementss has increased by 20%, from 1638 to 1979. Similarly, the number of measurements of heterotrophic respiration has increased from 53 measurements when Bond-Lamberty et al (2004) originally derived Eq. (4) to 362 measurements in the most recent SRDB version. To test the applicability of the original parameters, we also implemented the bottom-up approach at the increased number of SRDB location-years available since Hashimoto et al (2015), i.e. all




datapoints in SRDB v20170208. Consistent with the original study, for SRDB experiments for which the observed annual average was determined over a range of years, we used only the middle year in the range. We compared simulated to observed annual $R_s$ and $R_h$ for both the case of the maximum a posteriori parameters from the original Hashimoto et al (2015) study and for a set of updated model parameters determined by a non-linear least-square fit. For the updated parameters, the coefficients

of the $R_h$-$R_s$ relationship are also optimized. Because the updated parameters did not perform significantly better (see Sec. 3.2), the original parameters were used in the rest of this study.

No uncertainty was considered in Hashimoto et al. (2015). To determine the uncertainty of the bottom-up estimates, we tested them against SRDB observations. Measurements in the SRDB are highly concentrated in the mid-latitudes – 74% of $R_h$ measurements and 78% of $R_s$ measurements were made at a latitude greater than 30 ºN. The uncertainty of the bottom-up

estimates is therefore likely to exhibit significant spatial and temporal variability due to sampling error alone, on top of errors related to the imposed functional form and its parameterization. To find one or more covariates between the expected uncertainty of $R_h$ and other factors, the errors associated with bottom-up implements at the SRDB sites were linearly regressed against the following possible predictors: latitude, longitude, mean and standard deviation of precipitation, mean and standard deviation of temperature, and mean predicted $R_h$. Several non-linear functions of latitude were also tested. Of these, the mean

predicted $R_h$ and latitude were chosen as predictors because they had the greatest explanatory power (R = 0.23 when used in combination). Adding more predictor variables does not further increase the adjusted $R^2$.

In order to determine a spatio-temporally variable uncertainty range we calculated the 25th and 75th percentile of all 362 $R_h$ errors associated with using the bottom-up model. These formed a baseline globally averaged confidence interval $\delta_{base}$ that was then modified linearly based on the modelled $R_h$ and latitude (consistent with the linear regression tests mentioned

above):

$$\delta_i = \delta_{base,i}\left(\frac{\gamma_1 + \gamma_2 R_h^{bu} + \gamma_3 lat}{\gamma_4}\right), \hspace{4cm} (6)$$

where $i$ denotes either the 25th or 75th percent confidence interval, $R_h^{bu}$ is the predicted mean bottom-up heterotrophic respiration rate, $lat$ is the pixel latitude, $\gamma_{1-3}$ are regression parameters and $\gamma_4$ is the mean error of the bottom-up method across the SRDB dataset. Although the amount of variability in error captured using this method (R = 0.23) is still extremely low, no

alternative ways of capturing the expected spatio-temporal variability in bottom-up $R_h$ uncertainty exist, and poorly accounting for this variability is still expected to be more useful than not accounting for it at all.

## 2.3 Comparison Analyses

We compared the mean and uncertainty estimates of the top-down and bottom-up annual $R_h$ across latitudes. Because

no bottom-up estimates of the seasonal cycle of $R_h$ are available, we further compared the seasonality of $R_h$ in different regions to the seasonality of $R_s$ from bottom-up estimates and $R_{eco}$ from the top-down estimates. Pixels are seasonally aggregated for



simplicity and plotting and to reduce noise from the propagation of atmospheric inversion uncertainty. In particular, we consider high-latitude regions (latitude greater than 55 °N/S), mid-latitudes (latitude between 30 and 55 °N/S), dry tropics (latitude < 30 °N/S and mean annual precipitation less than 1500 mm/yr), and wet tropics (latitude < 30 °N/S and mean annual precipitation greater than 1500 mm/yr). To calculate the uncertainty of the bottom-up $R_s$ estimates, a method analogous to that

used for determining the 25th-75th confidence interval of bottom-up $R_h$ was used.

## 3 Results

### 3.1 Annual average $R_h$ from top-down and bottom-up estimates

#### 3.1.1 Top-down $R_h$

The annual mean tropical $R_h$ is 450 $\pm$ 200 gC/m$^2$/yr. The spatial pattern of mean annual $R_h$ is similar to that of GPP,

(Fig. 3a, $R^2 = 0.97$, $p < 0.001$), as expected. More complex dynamics are revealed by considering the coefficient of variation (CV) of $R_h$ (e.g. temporal standard deviation divided by mean per grid cell, Fig. 3b). The CV does not closely follow known spatial patterns in biomes, GPP, turnover times, or other carbon parameters (e.g. (Anav et al., 2015; Bloom et al., 2016; Carvalhais et al., 2014; Hiederer & Kochy, 2011)), as it reflects a combination of all these factors. More information about the temporal variability of substrate availability (e.g. litter and soil organic matter) is needed to disentangle the climatic and

biogeophysical controls on $R_h$ dynamics. This is left for a future investigation. Note that the high CV values in semi-arid regions are likely due to the near-zero mean $R_h$ there.

Fig. 4 shows the results of the sensitivity analysis assuming a temporally variable CUE. The magnitude of the $R_h$ change resulting from a change in CUE depends on whether the seasonality of GPP aligns with $R_h$ and whether the changed CUE causes unrealistic flux combinations across any of the ensemble members. The difference in time-average $R_h$ is relatively

small - no more than 50 g C m$^{-2}$ yr$^{-1}$ for any pixel. Despite the change in seasonality of CUE, the temporal dynamics of the 36 months of estimated $R_h$ also remain relatively similar in the sensitivity analysis. More than 90% of pixels have an $R^2$ between the $R_h$ from constant CUE and the $R_h$ from seasonally variable CUE greater than 0.8. The largest difference in $R_h$ seasonality occurs in the wet tropics. In these regions, the average GPP is largest, and a change in CUE seasonality corresponds to the greatest absolute change in $R_a$.

Fig. 5 maps the relative contributions of uncertainty in NEP, GPP, and CUE to $R_h$ as calculated by consecutively re-calculating Rh assuming in each case that all but one of the three datasets have zero uncertainty. The uncertainty in GPP is the dominant source of uncertainty in $R_h$ across most of the globe, except in parts of the Amazon. Consistent with the CUE sensitivity analysis (Fig 4), the contribution of CUE to the $R_h$ uncertainty is greatest in the tropics. In many high-latitude regions, NEP also contributes significantly to the overall $R_h$ uncertainty. Overall, future efforts to improve top-down

approaches for $R_h$ estimation would likely benefit most from reduced uncertainty in remotely sensed GPP estimates.



### 3.1.2 Bottom-up $R_h$

The performance of the bottom-up approach at SRDB sites for both $R_s$ and $R_h$ is shown in Fig. 6. The influence of latitude on modelled $R_h$ is stronger than on observed $R_h$ (since the color patterns in Fig. 6 are largely horizontal). The uncertainties of the bottom-up method are high. Indeed, for both the bottom-up $R_s$ and $R_h$, the root-mean-square error (RMSE)

(421 g C m$^{-2}$ yr$^{-1}$ for $R_s$, 306 g C m$^{-2}$ yr$^{-1}$ for $R_h$) is only less than 15% lower than the RMSE for a model that simply predicted the average observed respiration value everywhere (RMSE = 488 g C m$^{-2}$ yr$^{-1}$ for $R_s$, 333 g C m$^{-2}$ yr$^{-1}$ for $R_h$). The $R_s$ RMSE = 421 g C m$^{-2}$ yr$^{-1}$ is also higher than the 376 g C m$^{-2}$ yr$^{-1}$ RMSE value reported by Hashimoto et al. (2015) when their equation was applied to a smaller subset of the current SRDB dataset. The performance of the bottom-up model may be even worse on a cross-validation dataset that is entirely independent.

To test whether the bottom-up model can be improved, its parameters were optimized using a non-linear least squares fit. The resulting values ($F$ = 1.30 gC m$^{-2}$ dy$^{-1}$, $a$ = 0.0565 °C$^{-1}$, $b$ = 0 °C$^{-2}$, $\alpha$ = 9.8, $K$ = 0.0008 cm mo$^{-1}$, $c$ = 0.92, and $d$ = 0.75) were of a similar magnitude as the original parameters ($F$ = 1.68 gC m$^{-2}$ dy$^{-1}$, $a$ = 0.0528 °C$^{-1}$, $b$ = 0.000628 °C$^{-2}$, $\alpha$ = 0.98, $K$ = 1.20 cm mo$^{-1}$, $c$ = 1.22, and $d$ = 0.73), for all values except $K$ and $\alpha$, the two parameters controlling the relationship between precipitation and $R_h$. This suggests that precipitation is among the most uncertain controls of $R_h$, consistent with the fact that

moisture limitations on $R_h$ are mediated by soil water potential rather than precipitation. However, because using the optimized parameters led to only a 3% reduction in RMSE (from 306 g C m$^{-2}$ yr$^{-1}$ to 294 g C m$^{-2}$ yr$^{-1}$, Fig. S2), the original parameters were used elsewhere in the manuscript. Several constraints and alternative initial conditions were tested for fitting, but these did not lead to a better-performing fit (not shown). Some compensation between parameters is likely occurring when fitting to observations, reducing the quality of the fit.

In the absence of additional information about the bottom-up model uncertainty, the SRDB implementation and the associated errors were also used to determine a model for the uncertainty of the global bottom-up estimates. As shown in Fig. 7, the $R_h$ experiments in the SRDB over-represent mid-latitudes but under-represent low and high latitudes relative to the distribution of global land area. This can also be seen visually in a map of the experimental locations (Fig. S3). As a result, pixels with low $R_h$ (which are typical in the high-latitudes) are underrepresented in the SRDB, such that the bottom-up model

has greater uncertainty there. These factors are accounted for by the dynamic uncertainty model in Eq. (5).

### 3.1.3 Comparison

The top-down and bottom-up estimates and their uncertainties are compared in Fig. 8. Global maps of the two $R_h$ estimates are also shown in Fig. S4. Except in boreal regions and in Australia, the top-down estimates are greater than the

bottom-up estimates. This is reflected in their global averages, with mean $R_h$ rates of 452 g C m$^{-2}$ yr$^{-1}$ for top-down vs. 353 g C m$^{-2}$ yr$^{-1}$ for bottom-up estimates (43.6 Pg C yr$^{-1}$ and 33.4 Pg C yr$^{-1}$ respectively, summed across the globe). The highest magnitude fluxes are in the low-latitude tropics, consistent with findings for $R_s$ by Zhao et al. (2017), and the monotonic $R_h$ -



$R_s$ relationship in Eq. (5). The difference between the two estimates is also largest in this region - top-down estimates are an average of 281 g C m$^{-2}$ yr$^{-1}$ larger than bottom-up ones between 30 ºS and 30 ºN, but are only 10 g C m$^{-2}$ yr$^{-1}$ larger than bottom-up estimates between 30 and 45 ºN/ºS. When compared against SRDB observations (Fig. 6b), the bottom-up estimates were 500-2000 g C m$^{-2}$ yr$^{-1}$ or more lower than observations at several low-latitude sites, suggesting the bottom-up estimates

may be underrepresenting $R_h$ across the region. The tropics is also the region where the relative uncertainties in both top-down (57% median relative 25-75$^{th}$ confidence interval width) and bottom-up (76% median relative 25-75$^{th}$ confidence interval width) estimates are highest. For the bottom-up estimation, this is due to a lack of representative in situ observations, while for the top-down estimates, this is likely driven by uncertainties in NEP from atmospheric diffusion and satellite sampling in the atmopsheric inversions (Liu et al., 2014) and GPP (Parazoo et al., 2014). Remarkably, although uncertainty estimates for

both the bottom-up and top-down approaches were conservative, the two estimates are so different at low latitudes that there is almost no overlap in their uncertainty ranges.

The greatest overlap between the two datasets and their uncertainty range occurs between 30 and 50 degrees North, where more than 48% of SRDB observations fall and the bottom-up estimates are likely the most reliable. At high-latitudes (above º55 N), the top-down uncertainty narrows but the bottom-up uncertainty does not. In this region, bottom-up

uncertainties are about 30% greater than the uncertainties of the top-down $R_h$.

## 3.2 Seasonal cycle of respiration components

The bottom-up estimates only provide $R_h$ at annual timescales. To gain insight into the realism of the seasonal cycle of the top-down $R_h$ estimates, they are compared to the seasonal cycle of bottom-up $R_s$ and top-down $R_{eco}$ in several regions

in Fig. 9. Consistent with the low values of bottom-up $R_s$ (Sec 3.1.3), the top-down $R_s$ are not much lower than $R_h$. There is significant overlap between the uncertainty ranges of both in many region-month combinations, despite the fact that true $R_h$ is always lower than (or equal to) $R_s$ due to the occurrence of belowground autotrophic respiration. Indeed, the bottom-up $R_s$ and top-down $R_h$ nearly overlap in the period January-March in the wet tropics. Remarkably, during boreal winter at high latitudes, the top-down $R_{eco}$, $R_s$, and $R_h$ all agree. This is likely because the constant CUE assumption assumes that $R_a$ is near-

zero in boreal winter when GPP shuts down, which may not be realistic. Previous studies have found that winter-time $R_s$ can provide as much as 20% or more of total boreal soil $CO_2$ fluxes (see overview in Hobbie et al., 2000), but only 5.2% of bottom-up estimated $R_s$ and 8.8% of top-down estimated $R_h$ here occurs between December and February. In the dry tropics, the seasonal cycle of top-down $R_h$ is remarkably flat, and flatter than that of bottom-up $R_s$. This could be explained by the fact that temperature, moisture, and substrate variabilities do not vary the same way across the seasons and may partially compensate

for one another. However, more research is needed to determine what controls dry tropical variations in $R_h$ and a detailed investigation of this issue is beyond the scope of this paper.

The ratio of estimated $R_h$ to $R_{eco}$ spans between close to 1 in high-latitude winters and 0.4 in the wet tropics. Similarly, the ratio of $R_h$ to $R_s$ varies from 0.75 to 0.94 for different month-region combinations.



## 4 Discussion

### 4.1 Uncertainties in top-down and bottom-up approaches are both uncertain

Top-down estimates of $R_h$ are 30% higher, on average, than bottom-up estimates. At low-latitudes, the top-down estimates of $R_h$ are so much larger than the bottom-up ones that there is almost no overlap between the 25th-75th uncertainty

intervals, despite efforts to create conservative uncertainty intervals in each case. Consistent with these results, the bottom-up $R_h$ were previously shown to be biased low relative to models from the Climate Model Intercomparison Project 5 (CMIP5) (Taylor et al., 2012) in the low-latitudes, though it is unclear whether this is because CMIP5 models are biased high or because the bottom-up estimates are biased low relative to true $R_h$ (Hashimoto et al., 2015, Fig.. 10). Zhou et al. (2009) found that attributing a globally uniform $Q_{10}$ value decreases model-simulated average $R_h$ by 40%, and a similar dynamic may be causing

the bottom-up $R_h$ estimates to be too low. It should also be noted that the global average $R_s$ estimates of the bottom-up approach are 10-20 Pg lower than the six other estimates of global $R_s$ published in the last decade (Bond-Lamberty, 2018), and that a lower bottom-up $R_s$ leads to a lower bottom-up $R_h$.

The top-down and bottom-up approaches to estimation of $R_h$ have complementary strengths and weakness, as detailed in Table 1. Top-down estimates are indirect, and errors and uncertainties in any of the source datasets can propagate to errors

and uncertainties in the retrieved $R_h$. These include the assumption of a temporally constant CUE, which among others, can lead to unrealistically low $R_h$ in boreal winters. Additional uncertainties also include, for example, choices made in the atmospheric inversion (Peylin et al., 2013) or the retrieval of SIF and its scaling to GPP (e.g. whether a constant set of values is used, or whether this scaling is dynamic as in the Parazoo et al (2014) data used here). GPP is the most uncertain of the input fluxes (Fig. 5). Despite their uncertainties, the top-down estimates are globally representative. By contrast, bottom-up

upscaling starts with more accurate, direct observations of $R_h$, but suffers from a lack of representativeness: they are often temporal snapshots covering only a single or few years at a given site, with the time period observed varying dramatically between sites. More importantly, they under-represent boreal and tropical regions, and may over- or under-sample disturbed sites in different regions. While the uncertainties of the remote sensing datasets used for top-down estimation vary in space, remote-sensing based estimates of vegetation properties such as photosynthesis and plant traits (Schimel et al., 2015) and

biomass (Saatchi et al., 2015) have previously been argued to contain significantly lower sampling errors than bottom-up estimates. A similar dynamic is at play for $R_h$.

For the bottom-up approach, the errors associated with sampling bias are likely also exacerbated by the uncertainty in parametrizing a single functional model and the difficulty of parameter optimization. When the model parameters were re-fit on a version of the SRDB that was slightly expanded from that used in Hashimoto et al. (2015), the precipitation-sensitivity

parameterization changed dramatically, while the error statistics remained similar, suggesting possible overfitting. Furthermore, even comparing against an SRDB dataset that was similar to that used to derive the original parameters, the bottom-up approach barely had improved error statistics (RMSE of 306 g C $m^{-2}$ $yr^{-1}$) relative to a model that simply ignores



spatial variations and everywhere assigns the same value (RMSE of 333 g C m$^{-2}$ yr$^{-1}$). Such results suggest a structural problem with the underlying modelling approach (no good parameters exist), but also call into question whether currently used parameters are truly optimal given the model structure. In a recent study, machine-learning based approaches for estimating $R_s$ were able to explain 60-70% of the $R_s$ variability (Zhao et al., 2017), considerably more than the 35% variability explained

in this study using the Hashimoto et al. (2015) approach. If the robustness of machine learning based bottom-up upscaling methods can be further established, they may form a path forward for improved fidelity of bottom-up estimation of $R_h$, and for allowing estimation of $R_h$ at a temporal resolution finer than the current annual timescales. However, the number of $R_h$ observations in the SRDB - and presumably the literature as a whole - is 5 times smaller than the number of $R_s$ sites. Thus, additional measurements of $R_h$ are needed for this approach, and they must include under-sampled areas. This is unlikely to

be possible in the foreseeable future.

Despite the complementary sources of uncertainties in both top-down and bottom-up $R_h$ estimates, the strong overlap between the two estimates and their uncertainty ranges in latitude range 30-50ºN (the same latitude range where SRDB observations are most common, Fig. 7) is encouraging. Indeed, if the uncertainty of top-down estimates can be reduced, they could be used to benchmark or help parameterize models similar to those used in bottom-up $R_h$ estimation, allowing creation

of a longer record than may be possible with top-down observational data alone.

## 4.2 General applicability of the carbon balance inversion method

This paper introduced a new method for top-down estimation of $R_h$ by calculating it as the residual of the carbon balance. The propagation of uncertainty under realism constraints (in the form of the correct sign on each of the respiration

components and GPP) is key to avoiding large errors because the NEP is of considerably smaller magnitude than the other terms in the carbon balance. In this paper, we used large-scale, regionally available estimates for the carbon balance components, including recently developed atmospheric inversion-based NBE and NEP estimates from CMS-Flux. However, this approach could also be applied at smaller scales, for example using regional scale atmospheric inversions. If the local carbon use efficiency can be determined, the method could also be applied at smaller spatial and temporal scales, such as to

data from eddy covariance towers. For example, constraints based on estimates of $R_h$ from a carbon balance inversion could be useful in upscaling chamber-based soil respiration measurements to the tower scale, which could help explain inconsistencies between tower and chamber measurements of respiration fluxes (Barba et al., 2017; Phillips et al., 2016).

## 4.3 Implications for carbon climate feedbacks

The response of terrestrial net carbon fluxes to climate changes are likely to feed back to future climate (Bodman et al., 2013), but the sign and magnitude of this feedback is highly uncertain (Friedlingstein et al., 2014). The tropics likely form a dominant control on global carbon-climate feedbacks (Cox et al., 2000; Schimel et al., 2015). However, in the period 2010-



2015, GPP explained less than 1/3$^{rd}$ of variations in tropical NEP, suggesting an important role for $R_a$ and $R_h$ in controlling net terrestrial carbon uptake and its climate sensitivity (Sellers et al., 2018). A recent modeling study also suggested that $R_h$ forms a dominant control on NBP at multi-decadal timescales (Zhang et al., 2018). Studies of climate-carbon feedbacks commonly consider either $R_{eco}$ or $R_s$, but in doing so they conflate two separate respiration components (total $R_a$ and $R_h$, or belowground $R_a$ and $R_h$, respectively), which have different biogeophysical controls and responses to climate. The large spatial and temporal variations in the ratio of top-down heterotrophic to $R_{eco}$ and $R_s$ in Fig. 8 act as a reminder that heterotrophic respiration should be studied separately from other respiration fluxes in this context. Indeed, data-driven estimation of $R_h$ can be particularly useful for validation of earth system models, as many of these do not even include $R_s$ as a separate variable.

The recent launch of TROPOMI, which has daily coverage and approximately 7 x 3.5 km pixel resolution, will greatly increase measurements of SIF, and hence GPP (Kohler et al., 2018). Increased data density from OCO-2 (Sun et al., 2017a) and in the future GeoCarb (Polonsky et al., 2014) should also provide better regional estimates of NEP. With these and other improvements to remote sensing-driven estimates of GPP and NEP, top-down estimation of $R_h$ may be a promising avenue to better understand the role of $R_h$ fluxes in carbon-climate feedbacks. However, because the assumption of constant CUE employed here has a particularly strong effect on the seasonal cycle of $R_h$ in the wet tropics (Fig. 4b) care should be taken in assessing how this assumption propagates to other studies of top-down $R_h$ variations. Nevertheless, temporal CUE variations in previous studies in the tropics have shown that seasonal variations in CUE are 0.05 or smaller (Rowland et al., 2014), less than assumed in the sensitivity analysis performed here. If more light can be shed on the drivers of variations in CUE, top-down estimation may be a promising approach for understanding or bounding the role of $R_h$ in carbon-climate feedbacks.

**Data availability**

Data are available at https://github.com/agkonings/ReHet and http://cmsflux.jpl.nasa.gov. Upon acceptance of the manuscript, the heterotrophic respiration data will be further deposited with a DOI.

**Author Contributions**

AGK, AAB, and DSS conceived of the idea, and AGK, AAB and KWB designed the research. AGK. performed the research. AAB., JL, NCP, and KWB. contributed remote sensing datasets. AGK wrote the first draft of the manuscript and all authors edited the manuscript and contributed to the interpretation of results.

**Competing Interests**

The authors declare that they have no conflict of interest



## Acknowledgements

The research was partially carried out at the Jet Propulsion Laboratory, California Institute of Technology, under a contract with the National Aeronautics and Space Administration. KWB, JL, and AAB acknowledge the support of NASA NNH14ZDA001N-CMS. AGK, AAB, JL, DS, and KWB also acknowledge the support of NASA NNH16ZDA001N-IDS.

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

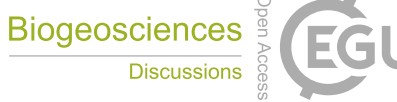

**Tables**

| | Top-down | Bottom-up |
|---|---|---|
| **Advantages** | Inherently global | Based on direct, high-resolution measurements |
| **Disadvantages** | Uncertainty in constant CUE assumption | Sparse, non-representative sampling |
| | Uncertainty in NEP and GPP data | Based on temporal snapshots in non-consecutive years |

**Table 1: Advantages and disadvantages of top-down vs. bottom-up estimation methods.**





**Figures**

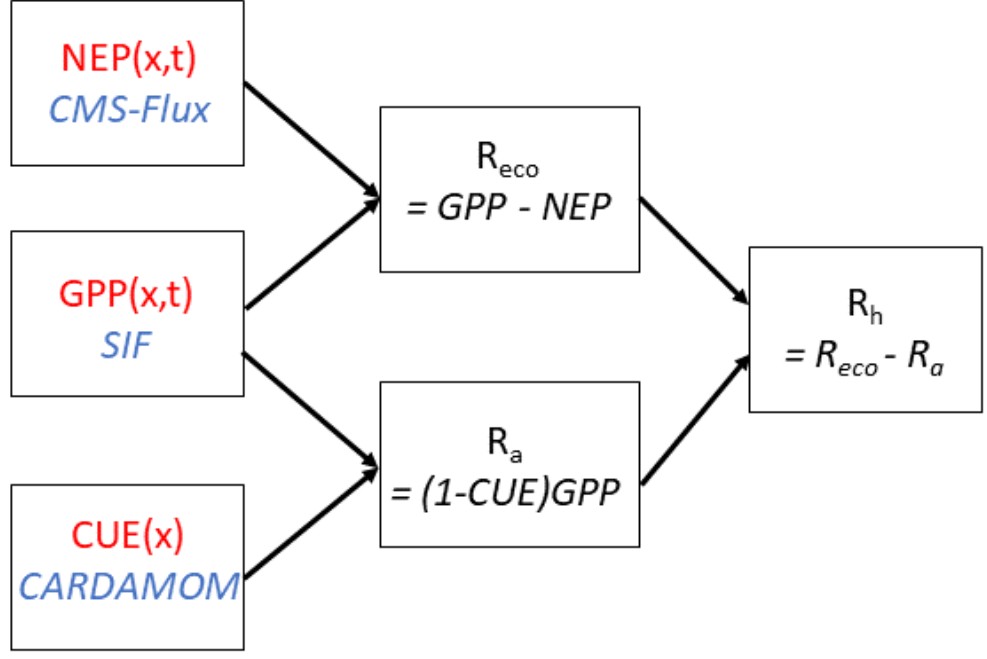

**Figure 1: Schematic diagram of process used to calculate heterotrophic respiration $R_{he}$. Input datasets are outlined in red, and data sources are described in blue italics. Arrows indicate one flux is used to calculate another. Data sources are described in detail in Section 2.1.2**

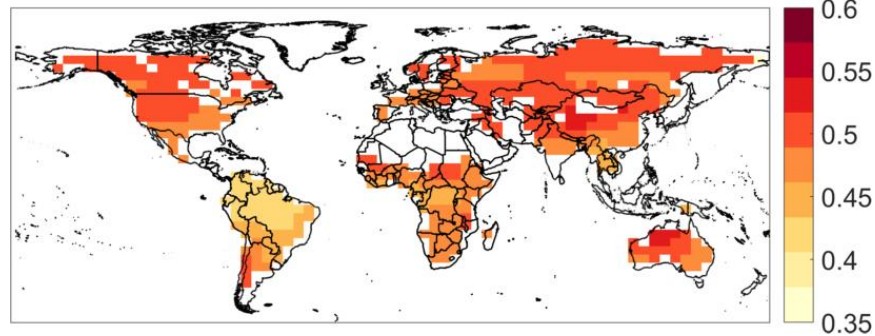

**Figure 2: Global variations in mean carbon use efficiency from CARDAMOM**




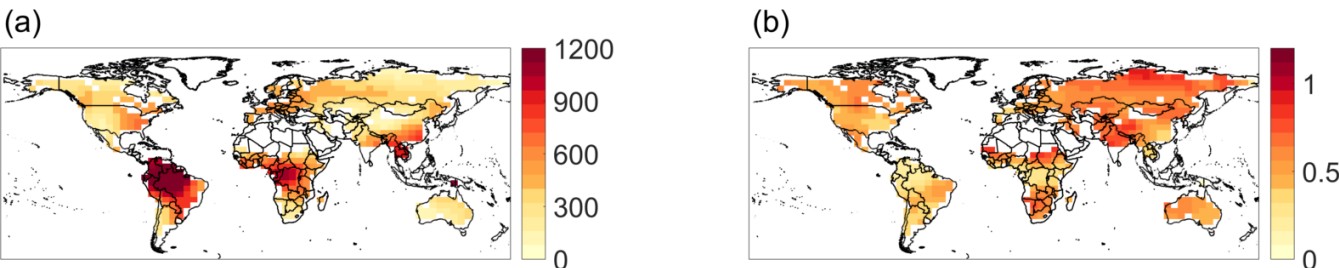

**Figure 3: Spatial variability in top-down $R_h$. Maps of a) mean $R_h$ [gC m$^2$ yr$^{-1}$] and right) temporal coefficient of variation of top-down $R_h$, calculated based on monthly data over 2010-2012.**

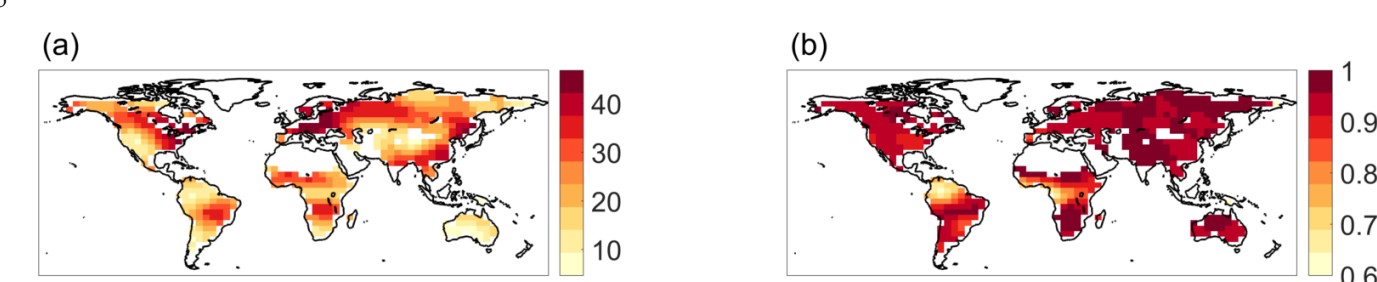

**Figure 4: Sensitivity analysis of constant CUE assumption. (left) Mean difference [gC m$^2$ yr$^{-1}$] between $R_h$ assuming constant CUE and $R_h$ assuming CUE varies temporally in a manner proportional to GPP and (right) coefficient of variation ($R^2$) between $R_h$ assuming constant CUE and $R_h$ assuming CUE varies temporally in a manner proportional to GPP.**

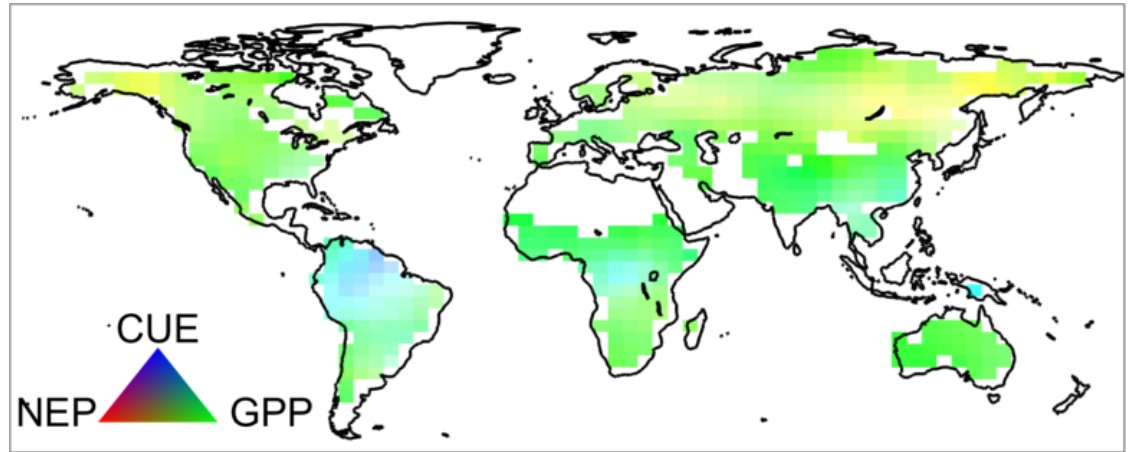

**Figure 5: RGB map of relative contributions to $R_h$ uncertainty in each of the input datasets, NEP (red), CUE (blue), and GPP (green).**



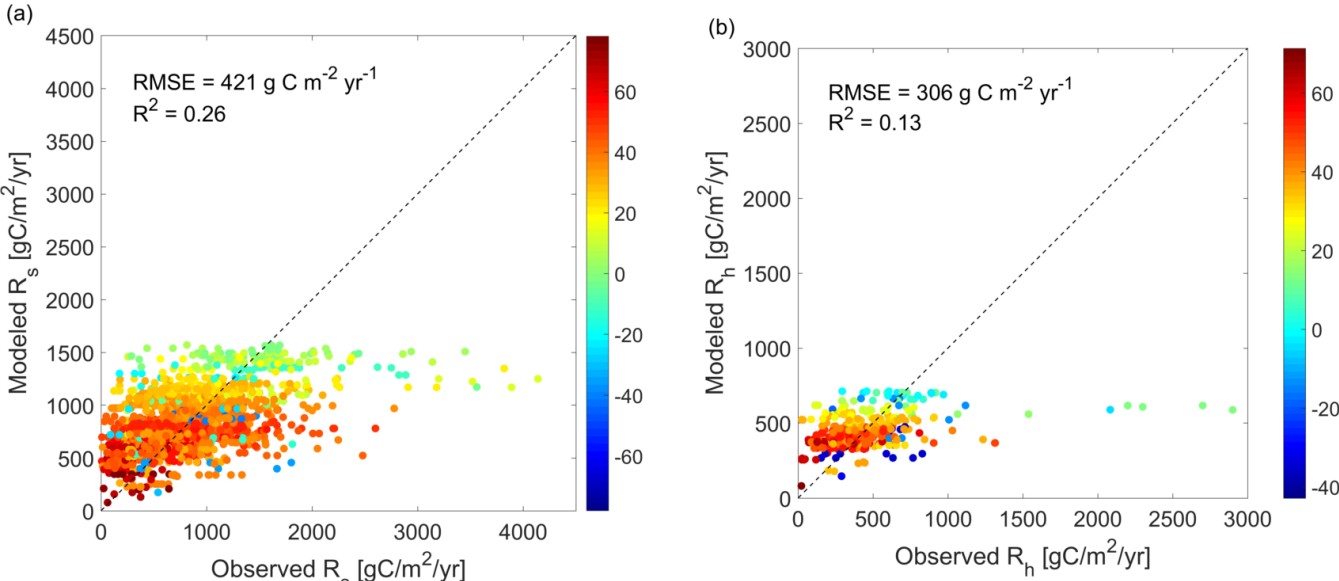

**Figure 6: Comparison of observed annual respiration terms at SRDB sites vs. bottom-up estimates at the same sites for (left) 1979 soil respiration sites and (right) 362 heterotrophic respiration sites. Each point denotes a single experiment and is colored by the experiment's latitude.**

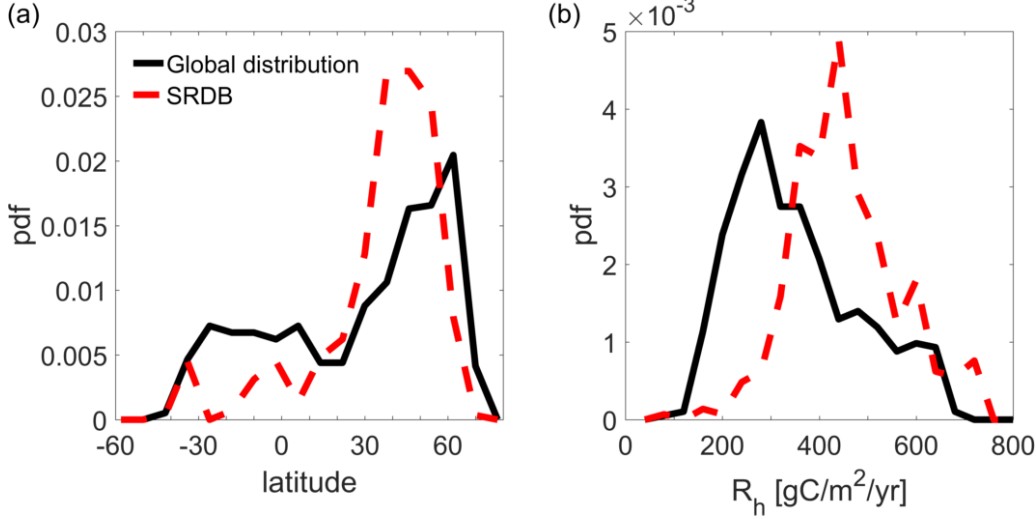

**Figure 7: Distribution of all SRDB experiments (red dashed lines) and global land points where top-down retrievals were possible in terms of (left) latitude and (right) bottom-up modelled $R_h$. Modelled $R_h$ rather than observed $R_h$ were used for the SRDB data in the comparison to isolate the differences due to the representativeness of the SRDB experiments relative to the entire global land area, and remove any possible effects of biases in modelled global values and observed SRDB values.**





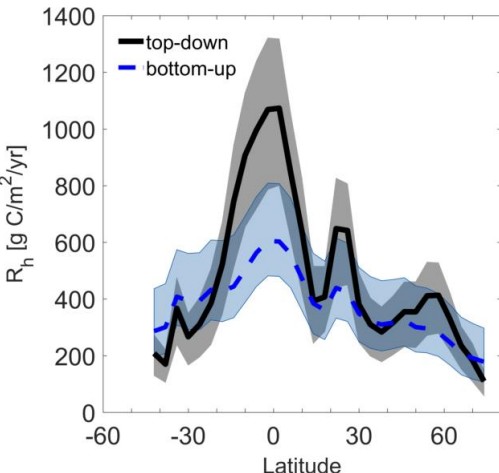

**Figure 8: Longitudinally-averaged $R_h$ as estimated from top-down (black solid line) and bottom-up (blue dashed line) estimates, respectively. Shaded areas represent the average 25th-75th uncertainty bars at each latitude.**

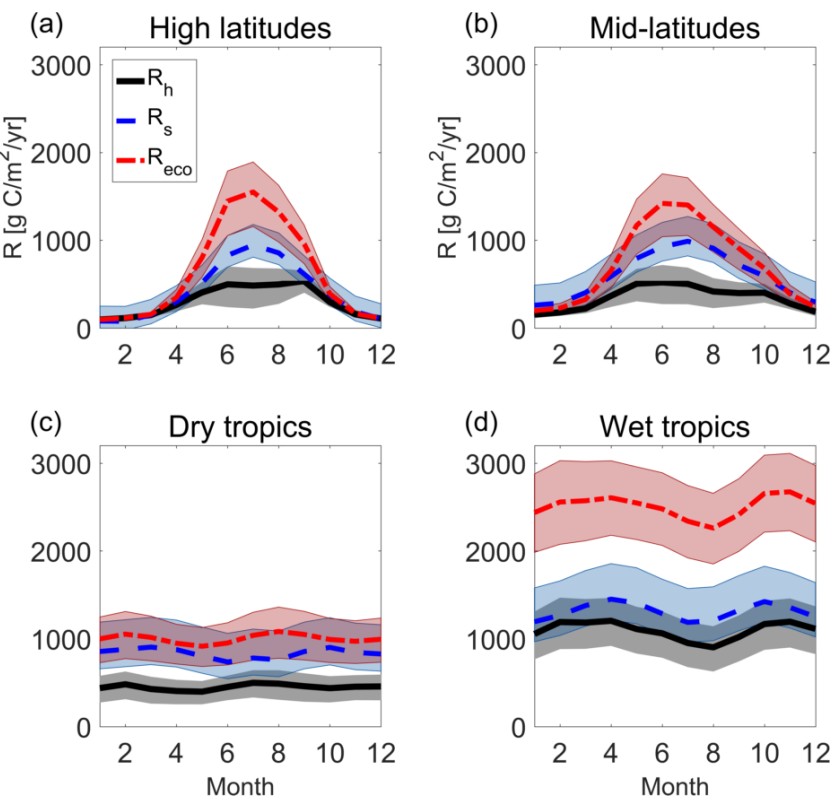

**Figure 9: Comparison between regionally and temporally averaged seasonal cycle of different respiration components: top-down $R_h$ (black solid line and area), bottom-up $R_s$ (blue dashed line and area), and top-down $R_{eco}$ (red dash-dotted line and area). Shaded areas represent the average 25th-75th uncertainty bars at each latitude. (top left) high latitudes (latitude > 55 N/S) (top right) mid-latitudes (30 N/S < latitude < 55 N/S), (bottom left) dry tropics (latitude < 30 N/S and mean annual precipitation < 1500 mm/yr), and**
10 **(bottom right) wet tropics (latitude < 30 N/S and mean annual precipitation > 1500 mm/yr).**