# Peer review of "Global, Satellite-Driven Estimates of Heterotrophic Respiration"

_Biogeosciences, 2018_

## Referee Comment (RC1) · Anonymous Referee #1 · 5 Dec 2018

Konings et al. provide globally distributed estimates of heterotrophic respiration (Rh), both from satellite based observations and from a bottom-up scaling of an empirical model. The satellite based estimates in particular are very novel, and are obtained by combining atmospheric inverse estimates of net ecosystem production (NEP) with global photosynthesis (GPP) estimates informed by solar induced fluorescence data, multiple vegetation models, and empirically upscaled estimates. To combine NEP and GPP to get an estimate of Rh, the authors need to estimate global variations in carbon use efficiency (CUE). The use estimates provided by CARDAMOM, a simple empirical model of carbon fluxes with parameters constrained by global observations. The final estimate of Rh is then taken as ðĺŚĚh =ðĺŘžðĺŚĊðĺŚĊ−ðĺŚĄðĺŘÿðĺŚĊ−(1−ðĺŘűðĺŚĹðĺŘÿ)ðĺŘžðĺŚĊðĺŚĊ.

The authors are to be commended for their heavily data-informed approach, which highlights the potential to use disparate observations to inform global estimates, not just test predictions. The results are interesting and the manuscript is very clearly written and no doubt will be of interest to the readers of Biogeosciences.

There are several limitations to the approach, however, and the estimates of Rh should be taken as a first pass of a promising approach rather than a reliable and informative quantification of the global distribution of Rh. As the authors note, the Rh estimates should not be used as a benchmark for other estimates, as their global quantified uncertainty is ∼50% of the mean flux. There are simply too many uncertainties, some quantified in this manuscript, and some not. Great caution should also be taken in using the approach to quantify trends over time.

The uncertainties stem primarily from the fact that both GPP and CUE are not known, but must be estimated themselves. Global GPP estimates vary a lot between approaches, and although the authors use an approach that combines SIF with DGVMs and upscaled GPP estimates, the relationship between SIF and GPP is poorly understood, and even the magnitude and spatial distribution of GPP has considerable uncertainty. The cited paper on which the GPP estimates are based, Parazoo et al. (2014) does a good job of assessing some of those uncertainties, but important sources of bias persist. For example, Parazoo et al. (2014) used the empirically upscaled GPP from Jung et al. 2012 to constrain the magnitude of GPP to roughly 120 PgC, but recent results of more updated empirical upscaling approaches from the FLUXCOM project (https://www.bgc-jena.mpg.de/geodb/projects/Data.php) show global GPP estimates vary from 108PgC (neural net based) to 125 PgC (Random forest based), each with an associated uncertainty of ∼8 PgC (standard deviation). It would be worth including an assessment of the contribution of this uncertainty to the global estimates reported here. The global estimates of CUE are also subject to large uncertainty, though the authors do a great job of assessing the impact on their results. In the absence of a global database of CUE and its seasonal variability, however, the uncertainty is difficult

to quantify accurately. It would be worth highlighting in the abstract what these uncertainties indicate regard research needs to improve this approach. Detailed comments:

The annual totals for GPP and NEP should be given in the methods section to allow the reader to assess their relationship with the annual total Rh. Page 2, line 20 Although the authors are correct that heterotrophic respiration is relatively unconstrained, the same cannot be said for ecosystem respiration, particularly at night. Eddy-covariance observations provide a direct observation of ecosystem respiration at night at 100's of sites around the world. Consider rewording.

Page 3, line 20: 'is calculated as'

Page 6, line 16: 'To reduce error, all visual maps are presented after applying a 3 pixel by 3 pixel moving average smoother.' This does not reduce error, unless the error is randomly distributed around zero. Do you have evidence that there is no systematic bias in spatial distribution of the CMS-Flux predictions?

Page 6, line 17: "The NEP is a small number that is the balance of many larger components, so small errors in NEP could lead to large compensating errors in Rh." This is not clear. A small error in NEP should have little effect on the derived Rh, as NEP itself has a small role in the calculation especially relative to GPP, which is a very large number, and CUE.

Page 6, line 25: the uncertainty in total annual GPP from the Parazoo et al (2014) paper does not consider methodological uncertainty (see differences between methods in FLUXCOM). How would this affect the results presented here.

Page 12, line 24: Most plant traits can not be estimated from space, and it is difficult if not impossible to properly characterize the uncertainty associated with estimates of photosynthesis from space as there are no observations of ecosystem photosynthesis. The authors should show some restraint when trying to argue that estimates of photosynthesis, plant traits and Rh from space contain significantly lower sampling errors

than bottom-up estimates. Also please clarify what you mean by sampling errors here and how sampling errors relate to total uncertainty.

Page 14, line 10: measurements of SIF and estimates of GPP. GPP is not measured by TROPOMI.

---

## Referee Comment (RC2) · Anonymous Referee #2 · 14 Feb 2019

Konings and colleagues aimed to derive global, satellite-driven estimates of heterotrophic respiration.

Here already lies the problem with the manuscript: Konings and colleagues focus too much on deriving the individual ecosystem fluxes that make up Rh top-down. GPP is derived from sun-induced fluorescence (top-down), but the uncertainty from using bottom-up estimates such as FLUXCOM is not evaluated. To my mind it should not matter if all fluxes that can be used to derive Rh top-down are also top-down estimates. Instead of using GPP from SIF also FLUXCOM-GPP (bottom-up) could be used – would that make a difference regarding spatial patterns?

For NEP the authors should discuss the effect of different products, for example Jena CarboScope NEP (http://www. bgc-jena.mpg.de/CarboScope/) or Chevallier et al.

[Figure]

(2010) or FLUXCOM (Zscheischler et al., 2017) (how problematic this may be).

On a similar note, one can get an estimate of Rh from CARDAMOM: this should be very much dictated by data. How does Rh from CARDAMOM compare to the satellite-driven estimates and Hashimoto's approach?

How different would global numbers be if NEP was 0 globally? Would spatial patterns change a lot? It seems like that due to the coarse NEP estimates you cannot achieve reasonable resolutions for Rh.

Overall, I cannot follow why we need such a coarse estimate of Rh. On page 14 line 7-8, the authors state that estimates of Rh can be helpful as a validation for ESMs. Using Ecosystem respiration as a validation would be enough to my mind. One evaluates temporal and spatial patterns of Reco to deduce if the representation of Ra and Rh can reproduce these patterns. In the approach presented here one ends up with partitioned Rh, but this heavily depends on the prescribed CUE.

Technical and other comments:

Page 7, line 13: Hashimoto et al. (2002), I think this should be 2015.

Figure 5: In the map there are yellow colors. In the RGB legend, however, yellow cannot be seen. Please correct.

References

Chevallier F, Ciais P, Conway TJ et al. (2010) CO 2 surface fluxes at grid point scale estimated from a global 21 year reanalysis of atmospheric measurements. 115.

Zscheischler J, Mahecha MD, Avitabile V et al. (2017) Reviews and syntheses: An empirical spatiotemporal description of the global surface–atmosphere carbon fluxes: opportunities and data limitations. Biogeosciences, 14, 3685-3703.

---

## Author Comment (AC1) · 7 Mar 2019

**Reviewer: Konings et al. provide globally distributed estimates of heterotrophic respiration (Rh), both from satellite based observations and from a bottom-up scaling of an empirical model. The satellite based estimates in particular are very novel, and are obtained by combining atmospheric inverse estimates of net ecosystem production (NEP) with global photosynthesis (GPP) estimates informed by solar induced fluorescence data, multiple vegetation models, and empirically upscaled estimates. To combine NEP and GPP to get an estimate of Rh, the authors need to estimate global variations in carbon use efficiency (CUE). The use estimates provided by CARDAMOM, a simple empirical model of carbon fluxes with parameters constrained by global observations. The final estimate of Rh is then taken as [sic] ðˈɪˈSEˇ h=ðˈɪ ˇ [...] The authors are to be commended for their heavily data-informed approach, which highlights the potential to use disparate observations to inform global estimates, not just test predictions. The results are interesting and the manuscript is very clearly written and no doubt will be of interest to the readers of Biogeosciences.**

*Response:* We are glad the reviewer believes our new satellite-based approach to estimating $R_h$ is very novel, and that the manuscript will be of interest to *Biogeosciences* readers.

**Reviewer: There are several limitations to the approach, however, and the estimates of Rh should be taken as a first pass of a promising approach rather than a reliable and informative quantification of the global distribution of Rh. As the authors note, the Rh estimates should not be used as a benchmark for other estimates, as their global quantified uncertainty is 50% of the mean flux. There are simply too many uncertainties, some quantified in this manuscript, and some not. Great caution should also be taken in using the approach to quantify trends over time.**

*Response:* We agree that the results of our approach remain uncertain, and tried to be clear about this. We are glad the reviewer also agrees that further work building on this first manuscript could help to improve the robustness of the method. In the revised version, we will add additional language to highlight the uncertainties of the approach, and its limitations for trend analysis and global benchmarking.

**Reviewer: The uncertainties stem primarily from the fact that both GPP and CUE are not known, but must be estimated themselves. Global GPP estimates vary a lot between approaches, and although the authors use an approach that combines SIF with DGVMs and upscaled GPP estimates, the relationship between SIF and GPP is poorly understood, and even the magnitude and spatial distribution of GPP has considerable uncertainty. The cited paper on which the GPP estimates are based, Parazoo et al. (2014) does a good job of assessing some of those uncertainties, but important sources of bias persist. For example, Parazoo et al. (2014) used the empirically upscaled GPP from Jung et al. 2012 to constrain the magnitude of GPP to roughly 120 PgC, but**

**recent results of more updated empirical upscaling approaches from the FLUXCOM project (https://www.bgc-jena.mpg.de/geodb/projects/Data.php) show global GPP estimates vary from 108PgC (neural net based) to 125 PgC (Random forest based), each with an associated uncertainty of _8 PgC (standard deviation). It would be worth including an assessment of the contribution of this uncertainty to the global estimates reported here. The global estimates of CUE are also subject to large uncertainty, though the authors do a great job of assessing the impact on their results. In the absence of a global database of CUE and its seasonal variability, however, the uncertainty is difficult to quantify accurately. It would be worth highlighting in the abstract what these uncertainties indicate regard research needs to improve this approach.**

*Response:* We agree that both the global CUE and GPP variability remain poorly understood, and that the Parazoo et al. (2014) approach is influenced by the quality of the Jung et al. (2012) estimates and TRENDY models. In the revised manuscript, we will include a sensitivity analysis that uses GPP estimated from FLUXCOM (the median across the three estimation methods) instead of the Parazoo et al. (2014) estimate used for the main calculations. We will also add text to the abstract regarding the dominant sources of uncertainties across the three fluxes and how our approach can be improved.

**Reviewer: Detailed comments:**

**The annual totals for GPP and NEP should be given in the methods section to allow the reader to assess their relationship with the annual total Rh.**

*Response:* Good point. We'll add these in the revised manuscript.

**Reviewer: Page 2, line 20 Although the authors are correct that heterotrophic respiration is relatively unconstrained, the same cannot be said for ecosystem respiration, particularly at night. Eddy-covariance observations provide a direct observation of ecosystem respiration at night at 100's of sites around the world. Consider rewording.**

*Response:* We will rephrase in the revised manuscript to mention eddy covariance sites as a widespread constraint on NEE (and nighttime respiration). However, we note that even the observational network of eddy covariance sites is not fully representative, with particularly strong biases in among others, regions of high topography and the Southern Hemisphere. For example, of the 225 sites used to constrain the FLUXCOM effort, only 17 are in the Southern Hemisphere.

**Reviewer: Page 3, line 20: 'is calculated as'**

*Response:* We'll fix this in the revised manuscript, thanks.

**Reviewer: Page 6, line 16: 'To reduce error, all visual maps are presented after applying a 3 pixel by 3 pixel moving average smoother.' This does not reduce error, unless the error is randomly distributed around zero. Do you have evidence that there is no systematic bias in spatial distribution of the CMS-Flux predictions?**

*Response:* In the revised version of the abstract, we will clarify that this moving average smoother was applied to reduce the random component of the error only. This can be accomplished simply by changing the quoted text to "to reduce the random component of the error, all visual maps are applied…". This 3x3 moving average window was also used by Liu et al, 2017.

Reference:

Liu et al. (2017): Detecting drought impact on terrestrial biosphere carbon fluxes over contiguous US with satellite observations. Biogeosciences Discussions, in review. https://doi.org/10.5194/bg-2019-37

**Reviewer: Page 6, line 17: "The NEP is a small number that is the balance of many larger components, so small errors in NEP could lead to large compensating errors in Rh." This is not clear. A small error in NEP should have little effect on the derived Rh, as NEP itself has a small role in the calculation especially relative to GPP, which is a very large number, and CUE.**

*Response:* This was indeed unclear – we meant small errors in NEP in combination with errors in GPP. We'll remove this in the revised manuscript, thanks for pointing it out.

**Reviewer: Page 6, line 25: the uncertainty in total annual GPP from the Parazoo et al (2014) paper does not consider methodological uncertainty (see differences between methods in FLUXCOM). How would this affect the results presented here.**

*Response:* A full assessment of the methodological uncertainty in any given GPP estimate is difficult without perfect knowledge of the GPP. For example, even the different methods in FLUXCOM do not capture all methodological uncertainty as they do not, for example, capture uncertainty related to possible missing input information (which may be part of the reason the uncertainty of the FLUXCOM products is actually lower than that of the Parazoo et al. (2014) estimates). Insofar as there is non-captured uncertainty in GPP, this would have a non-negligible effect on the Rh, as shown in Figure 5. We will further illustrate this in the new manuscript with a new sensitivity analysis that uses FLUXCOM GPP.

**Reviewer: Page 12, line 24: Most plant traits can not be estimated from space, and it is difficult if not impossible to properly characterize the uncertainty associated with estimates of photosynthesis from space as there are no observations of ecosystem photosynthesis. The authors should show some restraint when trying to argue that estimates of photosynthesis, plant traits and Rh from space contain significantly lower sampling errors than bottom-up estimates. Also please clarify what you mean by sampling errors here and how sampling errors relate to total uncertainty.**

*Response:* By sampling errors, we mean errors associated with the fact that in situ observations may not be in locations that are representative of environmental conditions across the globe – for example, because they are underrepresented in the tropics, or because they occur more frequently in disturbed areas than ecosystems as a whole, or because they undersample regions of high topography, etc. Because they sample across the globe, spaceborne remote sensing estimates would be expected to have significantly lower representativeness errors, although we agree with the reviewer that some errors may remain due to e.g. reductions in accuracy correlated with cloud cover. We also agree that remote sensing of photosynthesis (and net carbon fluxes) still has significant sources of error. We did not mean to imply in this section that any top-down estimate of an environmental variable will always have lower overall uncertainties. We will rephrase the text in the revised manuscript to clarify all of these issues, including clarifying our use of the term 'sampling error' and removing the phrase 'plant traits'.

**Reviewer: Page 14, line 10: measurements of SIF and estimates of GPP. GPP is not measured by TROPOMI.**

*Response:* Thanks, we'll fix this.

---

## Author Comment (AC2) · 7 Mar 2019

We thank the reviewer for their helpful comments, and agree that more detailed sensitivity analyses can clarify the uncertainty of the method. A point-by-point response is included below.

**Konings and colleagues aimed to derive global, satellite-driven estimates of het-erotrophic respiration.**

**Reviewer: Here already lies the problem with the manuscript: Konings and colleagues focus too much on deriving the individual ecosystem fluxes that make up Rh top-down. GPP is derived from sun-induced fluorescence (top-down), but the uncertainty from using bottom-up estimates such as FLUXCOM is not evaluated. To my mind it should not matter if all fluxes that can be used to derive Rh top-down are also top-down estimates. Instead of using GPP from SIF also FLUXCOM-GPP (bottom-up) could be used –would that make a difference regarding spatial patterns?**

*Response:* As we discuss in Sec 4.1, bottom-up estimates are generally based on sparse samples that are often not representatives. For example, FLUXCOM products are highly undersampled in the tropics. For example, of the 225 sites used to train FLUXCOM, only 17 were in the Southern Hemisphere, and of those 4 were in Australia. With such a low number of training data in the wet and dry tropics, the model is likely overfitting in the climatic conditions of these regions. This is why we chose the more top-down approach here. We explicitly contrast this to the best-available bottom-up dataset for $R_h$ and show the uncertainties are comparable.

Nevertheless, we will include a sensitivity analysis using FLUXCOM GPP in the revised manuscript. While using FLUXCOM GPP affects the spatial patterns of the resulting $R_h$ more than the CUE sensitivity analysis in Fig. 4 of the current manuscript (which is also consistent with the uncertainty analysis in Fig. 5 of the current manuscript), the difference in the spatial difference of $R_h$ between different GPP assumptions is still less than the difference between the top-down and bottom-up products. The absolute differences are largest in the mid-latitudes and boreal regions.

**Reviewer: For NEP the authors should discuss the effect of different products, for example Jena CarboScope NEP (http://www. bgc-jena.mpg.de/CarboScope/) or Chevallier et al. (2010) or FLUXCOM (Zscheischler et al., 2017) (how problematic this may be).**

*Response:* An exhaustive discussion of propagated effects of different NEP and GPP product combinations is beyond the scope of this manuscript. However, in the revised version, we will add an uncertainty analysis for NEP with a value of zero everywhere, as suggested by this reviewer in a later comment. Doing so changes the spatial distribution of mean $R_h$ less than changing the GPP assumptions, but does create a greater (normalized) root-mean-square-difference with the baseline $R_h$. We will discuss these results in the revised manuscript.

**Reviewer: On a similar note, one can get an estimate of Rh from CARDAMOM: this should be very much dictated by data. How does Rh from CARDAMOM compare to the satellite-driven estimates and Hashimoto's approach?**

*Response:* While CARDAMOM uses model-data fusion to incorporate information from remote sensing products, its soil carbon pools and fluxes are much less well constrained than the aboveground carbon pools and fluxes. The version of CARDAMOM used in this manuscript (that most consistent with the

published literature, e.g Bloom et al PNAS 2016) predicts heterotrophic respiration only as a function of pixel-dependent base respiration rates, turnover time, and temperature, but does not account for water. As such, while CARDAMOM $R_h$ and our newly derived $R_h$ are largely consistent in the mid-latitudes and boreal regions, they actually have opposite seasonality in the dry tropics, where soil moisture limitations largely drive the seasonal cycle of $R_h$ and the CARDAMOM $R_h$ is unrealistic.

We emphasize that this does not mean CARDAMOM CUE cannot be used in our top-down method. As we have previously discussed in the manuscript, the allocation fractions that influence the CUE are particularly well-constrained. Indeed, the spatial variability in CARDAMOM is consistent with a recent meta-analysis compiling carbon use efficiency across 188 sites that has been submitted to Biogeosciences Discussions since the time we originally submitted our manuscript (Tang et al, in review). We will include this reference in the revised manuscript. We have also since updated CARDAMOM to account for the effects of soil water limitations on heterotrophic respiration, with little qualitative effect on the resulting global vegetation CUE maps, but have not included this set of maps in the current work because they have not been published elsewhere and the validity of changes to CARDAMOM is beyond the scope of this research.

Reference:

Tang et al (2019): Global variability of carbon use efficiency in terrestrial ecosystems. *Biogeosciences Discussions*, in review. https://doi.org/10.5194/bg-2019-37

**Reviewer: How different would global numbers be if NEP was 0 globally? Would spatial patterns change a lot? It seems like that due to the coarse NEP estimates you cannot achieve reasonable resolutions for Rh.**

*Response:* The 4° by 5° resolution is not much coarser than other atmospheric inversions. For example, the Jena CarboScope has the same resolution, and the Chevallier et al (2010) data mentioned by the reviewer in an earlier comment have a resolution of 3.75° by 2°. As we discuss in Section 4.3, these resolutions may also become finer in the future when the spatial resolution of remotely sensed xCO2 measurements improves with OCO-3 and GeoCarb measurements. Furthermore, as we discussion in Section 4.2, one could also use the approach presented in this paper using higher-resolution regional-scale (instead of global-scale) atmospheric inversions for particular applications where resolution is a significant concern.

In the revised manuscript, we will include a sensitivity analysis for NEP with a global value of zero. As we mentioned above, this changes the spatial distribution of mean $R_h$ less than changing the GPP assumptions, but does create a greater (normalized) root-mean-square-difference with the baseline $R_h$. This suggests the temporal variability of $R_h$ is affected by the temporal variability of NEP, and that inclusion of an accurate atmospheric inversion can help constrain $R_h$.

**Reviewer: Overall, I cannot follow why we need such a coarse estimate of Rh. On page 14 line 7-8, the authors state that estimates of Rh can be helpful as a validation for ESMs. Using Ecosystem respiration as a validation would be enough to my mind. One evaluates temporal and spatial patterns of Reco to deduce if the representation of Ra and Rh can reproduce these patterns. In the approach presented here one ends up with partitioned Rh, but this heavily depends on the prescribed CUE.**

*Response:* We acknowledge that our method depends on the assumed constancy of the CUE, and have tried to be transparent about this uncertainty, including in the sensitivity analysis in Figure 4, and in the Discussion in Section 4.3, where we write:

> "However, because the assumption of constant CUE employed here has a particularly strong effect on the seasonal cycle of $R_h$ in the wet tropics (Fig. 4b) care should be taken in assessing how this assumption propagates to other studies of top-down $R_h$ variations"

In the revised manuscript, we will expand on this text to clarify that such care should be taken everywhere, not just in the tropics.

In terms of why even coarse top-down $R_h$ data can be useful: global $R_h$ remains a highly uncertain flux, as discussed in detail in Tian et al. (2015) and Bond-Lamberty et al. (2016). Furthermore, in the 4 years since it has been published, the bottom-up global Hashimoto et al. dataset (2015) has been cited 73 times. While not all of these citations focused on $R_h$, we show that the Hashimoto et al. (2015) approach is sensitive to the overfitting that occurred in that dataset for both $R_s$ and $R_h$. Furthermore, our manuscript introduces not just an alternative top-down dataset, but also a method that could be applied at a variety of scales and (at smaller scales) resolutions.

Lastly, as to why $R_h$ is preferable to $R_{eco}$ for validating Earth System Models: $R_a$ and $R_h$ can have differential sensitivity to drought and other climatic variations; see, for example, two papers cited below (Sun et al, 2019; Zhang et al, 2019) that have been published only in the last few months making this point, though of course there are others. Thus, while $R_{eco}$ can be used as an indirect constraint on $R_h$ and $R_a$, knowing only $R_{eco}$ is not enough to unambiguously determine which process representation in the models needs the most improvement, and to test possible alternatives (particularly since, for example, a change to the Ra formulation could affect how much carbon is left for allocation to foliar carbon pools, and thus eventually for $R_h$). As we recommend in Section 4, when using data derived from our proposed method, alternative CUE assumptions can be easily tested to ensure model evaluations are not affected. We will expand on this discussion point in the revised manuscript.

References

Bond-Lamberty, B., Epron, D., Harden, J., Harmon M.E., Hoffman, F., Kumar, J., McGuire, A.D., and R. Vargas (2016): Estimating heterotrophic respiration at large scales: challenges, approaches, and next steps. *Ecosphere* 7(6):e01380, doi.org/10.1002/ecs2.1380.

Sun, S., Lie, H., and S.X. Chang (2019): Drought differentially affects autotrophic and heterotrophic soil respiration rates and their temperature sensitivity. *Biology and Fertility of Soils.* doi.org/10.1007/s00374-019-01347-w

Tian, H., Lu, C., Yang, J., Banger, K., Huntzinger, D. N., Schwalm, C. R., Michalak, A. M., Cook, R., Ciais, P., Hayes, D., Huang, M., Ito, A., Jain, A. K., Lei, H., Mao, J., Pan, S., Post, W. M., Peng, S., Poulter, B., Ren, W., Ricciuto, D., Schaefer, K., Shi, X., Tao, B., Wang, W., Wei, Y., Yang, Q., Zhang, B. and Zeng, N. (2015): Global patterns and controls of soil organic carbon dynamics as simulated by multiple terrestrial biosphere models: Current status and future directions, *Glob. Biogeochem*. 10 Cycles, 29, 775–792, doi:10.1002/2014GB005021.

Zhang F., Quan Q., Ma, F., Tian D., Zhou Q., and S. Niu (2019): Differential responses of ecosystem carbon flux components to experimental precipitation gradient in an alpine meadow. *Functional Ecology* 1-12. doi.org/10.1111/1365-2435.13300

**Technical and other comments:**

**Page 7, line 13: Hashimoto et al. (2002), I think this should be 2015.**

*Response*: We will fix this, thank you

**Figure 5: In the map there are yellow colors. In the RGB legend, however, yellow cannot be seen. Please correct.**

*Response*: The yellow is between green and red on the bottom axis. We will make it more prominent in the revised version.

**References**

Chevallier F, Ciais P, Conway TJ et al. (2010) CO 2 surface fluxes at grid point scale estimated from a global 21 year reanalysis of atmospheric measurements. 115.

Zscheischler J, Mahecha MD, Avitabile V et al. (2017) Reviews and syntheses: An empirical spatiotemporal description of the global surface–atmosphere carbon fluxes: opportunities and data limitations. Biogeosciences, 14, 3685-3703.